# Thyroid Microcarcinoma in Pediatric Population in Romania

**DOI:** 10.3390/children8050422

**Published:** 2021-05-20

**Authors:** Andreea-Ioana Stefan, Andra Piciu, Maria Margareta Cosnarovici, Monica Dragomir, Romana Netea-Maier, Doina Piciu

**Affiliations:** 1Doctoral School “Iuliu Hațieganu”, University of Medicine and Pharmacy, 400012 Cluj-Napoca, Romania; andreea.stefan24@gmail.com (A.-I.S.); cosnarovici_maria@yahoo.com (M.M.C.); doina.piciu@gmail.com (D.P.); 2Department of Medical Oncology “Iuliu Hațieganu”, University of Medicine and Pharmacy, 400012 Cluj-Napoca, Romania; 3Institute of Oncology “Carol Davila”, University of Medicine and Pharmacy, 022328 Bucharest, Romania; md.dragomir@gmail.com; 4Department of Internal Medicine, Division of Endocrinology, Radboud University Medical Center, 6525 GA Nijmegen, The Netherlands; Romana.Netea-Maier@radboudumc.nl; 5Department of Endocrine Tumors and Nuclear Medicine, “Prof. Dr. Ion Chiricuţă” Institute of Oncology, 400012 Cluj-Napoca, Romania

**Keywords:** pediatric, thyroid cancer, microcarcinoma

## Abstract

Thyroid microcarcinoma in pediatric population in Romania Non-medullary thyroid cancer (TC) is the most common endocrine malignancy, with an increasing incidence in the recent years, due to the increase of the thyroid microcarcinoma. Thyroid microcarcinoma (mTC) is defined, according to WHO criteria, as ≤1 cm dimension thyroid carcinoma, being a rare disease in children population. In adults, the current guidelines recommend a limited surgical approach. In children, however, there are no specific guidelines for mTC. Due to the scarcity of these tumors, mTC in children have largely been understudied, to our knowledge with only one previous publication reporting on the outcomes of a large historic series of patients with mTC from the USA. In Romania, the incidence of TC is rising, one of the reason may be the effect of Chernobyl nuclear accident in the past and the iodine deficiency. The purpose of this study was to describe the characteristics and outcome of children diagnosed with mTC in Romania diagnosed from 1 January 2000 to 31 December 2018. During the study period we identified 77 cases of differentiated TC (papillary and follicular) and of these 20 cases (19.4%) were mTC. The mTC represented roughly one fifth of our nationwide pediatric population diagnosed in the last 20 years, the majority of cases being recorded in adolescents aged between 15–18 years. Although patients with apparently more unfavorable local phenotype were identified, this was not reflected in the outcome of the patients in terms of remission of the disease and survival. Our study illustrates the heterogeneity of the real-life practice with respect to the pediatric mTC, and underscores the need for carefully designed multicenter international studies, including larger cohorts of patients in order to provide the data required for establishing evidence based uniform protocols. The European Reference Networks (ERN), such as the ERN for Rare Endocrine Diseases (Endo-ERN) provides an ideal platform to initiate such collaborative studies.

## 1. Introduction

Nonmedullary thyroid cancer (TC) is the most common endocrine malignancy, with an increasing incidence in recent years, due to the increase of thyroid microcarcinoma [1].

Thyroid microcarcinoma (mTC) is defined, according to WHO criteria, as ≤1 cm dimension thyroid carcinoma [2]. A thyroid cancer diagnosis is based on clinical examination, thyroid ultrasound, and fine-needle aspiration. In the case of mTC, the diagnosis is often established incidentally following histopathological analysis of surgery performed for benign conditions such as Graves’ disease or multinodular goiter [3]. Most cases of mTC have a favorable evolution, with good overall survival, over 99%, as in most cases of differentiated TC [4].

In children, TC is rare [5,6]. In adults, the current guidelines recommend a limited surgical approach, which includes lobectomy, and does not require radioactive iodine ablation therapy, and some advocate active surveillance or local thermal ablation procedures for the nonaggressive cases [7,8,9,10]. In children, however, there are no specific guidelines for mTC [11]. Due to the scarcity of these tumors, mTC in children has largely been understudied to our knowledge, with only one previous publication reporting on the outcomes of a large historic series of patients with mTC from the USA [4].

In Romania, the incidence of TC is rising [12,13,14,15]. One of the reasons for this may be the effect of the Chernobyl nuclear accident in the past and the iodine deficiency, the Romanian population being classified in 2002 by the World Health Organization as having an average iodine deficiency, with an insufficient intake of additional iodine [16]. We have presented [15] a series of TC in the pediatric population diagnosed between 2000 and 2018 in one large regional center, the “Ion Chiricuta” Institute of Oncology Cluj-Napoca (IOCN), covering the central and western part of Romania, identifying mTC in 14.5% of cases. To zoom in and better characterize this mTC patient group, we decided to extend this cohort with the addition of another reference center in Romania—“Prof Dr. Alexandru Trestioreanu” Institute of Oncology Bucharest (IOB). The databases of these institutions represent almost 75% of the entire pediatric thyroid neoplastic pathology in Romania. 

The purpose of this study was to describe the characteristics and outcome of children diagnosed with mTC, all of them being papillary thyroid microcarcinoma (mPTC), in Romania using this database with pediatric TC cases diagnosed from 1 January 2000 to 31 December 2018. 

## 2. Patients and Method

### 2.1. Design of the Study and the Population

This retrospective observational study evaluated children aged between 0 and 18 years and 6 months diagnosed with mPTC, in the registry of the IOCN and IOB, in the period 01.01.2000-31.12.2018. Given that the development of TC precedes the diagnosis by at least a few months up to several years, we decided to set the upper age limit of inclusion in this study to 18 years and 6 months at the time of diagnosis.

The data collection was carried out retrospectively, from the medical records of the patients from the above institutions. The study was approved by the Ethics Committee of the Iuliu Hatieganu University of Medicine and Pharmacy Cluj-Napoca (no 58/11.02.2020), of the IOCN (No 167/05.02.2020), and of the IOB (No 2791/24.02.2020). All patients or legal representatives signed informed consent prior to inclusion. 

### 2.2. Study Variables

The following variables were collected: age at the time of diagnosis, sex, the date of diagnosis defined as the date of pathology report release, the duration of follow-up as the time elapsed from the date of diagnosis to the last medical visit noted in the medical file (years), clinical presentation and thyroid function status at diagnosis, histological type, and disease staging, treatment received, complications and outcome of treatment. Considering the ATA recommendation from 2015 [11], we also analyzed the age of diagnosis in order to evaluate the impact of pubertal hormonal status in this pathology. 

Hypoparathyroidism was defined as the need to administer calcium and vitamin D supplements postoperatively. If this lasted for more than 6 months after thyroidectomy this was regarded as permanent; otherwise, it was considered transient. Recurrent laryngeal nerve damage was defined according to the presence of hoarseness, followed by confirmation of vocal cord dysfunction at laryngoscopy or if the tumor invaded the recurrent laryngeal nerve. 

Complete remission was considered when there were no signs and symptoms suggestive for clinical, radiological, or serological disease defined as undetectable serum thyroglobulin (Tg) (according to the institutional cut-off level) after thyroxine withdrawal or after administration of recombinant human TSH (rTSH) at least 6 months after the last I-131 administration and negative antithyroglobulin (anti-Tg) antibodies. Biochemical incomplete remission was considered when there was no clinical or radiological evidence of the disease, but there was still detectable (stimulated or unstimulated) Tg. The persistent structural disease was defined as clinical and radiological evidence of disease. The recurrence of the disease was defined as new clinical, pathological, radiological, or biochemical evidence of the disease after the remission has been established before. In patients who had only a lobectomy, remission was based on the clinical and radiological criteria and a stable Tg level. The diagnostic and treatment protocols were the same in all three institutions with respect to indication and extent of surgery and follow-up but differed with respect to amount and frequency of I-131 activity administered and changed during the period covered by the study and according to the staging and recommendation of the moment.

Considering that during the timeframe of the study, the TNM staging changed several times, both the initial staging, in use at the time of diagnosis, and the restaging, made according to the eighth edition of 2017 TNM classification [17,18], were noted. Lymphovascular invasion was considered according to AJCC Cancer Staging Form Supplement [19]. This results in uniformity of staging according to the most recent recommendations of the American Thyroid Association (ATA), and they were classified according to the risk of relapse into one of three risk groups [11] Table 1.

In order to assess survival, the electronic platform of the national health insurance company (http://www.cnas.ro/page/verificare-asigurat.html, accessed on 25 January 2021) was questioned by the ID number of each patient.

## 3. Results

During the study period, we identified 77 cases of differentiated TC (papillary and follicular), and of these, 15 cases (19.4%) were mTC; all of them were papillary thyroid microcarcinoma. Most cases of mPTC were registered in 2014 (30%) (Figure 1). 

By 25.01.2021 (date of interrogation), all patients were alive. The median duration of follow-up was 4.7 years (0.9–16.4 years).

The clinical characteristics of the individual patients are presented in Table 2. The entire cohort of patients with mPTC included 12 girls (F) and 3 boys (M), with a median age of diagnosis of 16.2 years (10.5–18.1 years). The majority of cases were older than 14 years old (13 children); the rest of them had 10.5 and 12.2 years old. One case had an atypical localization in a thyroglossal cyst (6.6%). Thyroid hormone status at diagnosis was in most cases of euthyroidism (73%), one case (6.6%) had pre-existing hypothyroidism, and two cases had hyperthyroidism caused by Graves’s disease (13.3%). Family medical history was positive in one case (F, 14.7 years) had the brother with papillary TC. She was not previously known with a hereditary tumor syndrome. 

We were able to retrieve information about the clinical presentation of 10 cases (66.6%). All of these patients presented with structural changes in the cervical region (eight patients with goiter, two patients with other cervical lesions, which were revealed to be, in one case, a thyroglossal cyst, and in the other, a thyroid nodule) noticed by the parents of the patient. This was associated in two patients with compressive symptoms (dyspnea, difficulty swallowing). One patient also had cervical adenopathy at physical examination. 

All patients had a preoperative thyroid ultrasound (US), but only nine cases of US records were available for review. Of these, the US revealed a nodular goiter in six cases, and a thyroid macronodule in two cases. In one patient, the US also revealed cervical lymph nodes, which were highly suspicious of malignancy. This was the only patient in whom a suspicion of malignancy was raised preoperatively, based on the US; none of the patients had a fine-needle aspiration biopsy (FNAB), preoperatively, a fact that must be underlined as an important difference, compared with the strategy in adult patients. 

Of all the 15 patients with mPTC, only one had a suspicion of malignancy at diagnosis. In the remaining, the diagnosis of mPTC was made postoperatively, after surgery for an unrelated condition, which was a (multi)nodular goiter (in eight patients) or therapy-resistant Graves’s disease (in two patients). 

Enrollment in risk groups showed that 86% of cases are in the low-risk group (T1a), and only two cases (13.3%) are in the intermediate-risk group (based on either local invasion or presence of lymph nodes metastases) (T3/N1b). All cases were in stage I, and only one case had lymph node metastases, but no other distant metastases were recorded. The dimensions of the tumor ranged from 0.1 to 1 cm with 40% of them below 5 mm. Multifocality was found in 53% of the cases.

Among the two patients who had intermediate-risk tumors, one showing a T3N1bM0 tumor at the histological examination had a previously uninvestigated multinodular goiter, which had been present before the presentation and had pathologic lymph nodes both at the time of physical examination and the US. The second patient with an intermediate-risk tumor (T3NxMx) had Graves’s disease, with thyrotoxicosis insufficiently responding to thyrostatic treatment for two years, and who underwent thyroidectomy for this indication. In this patient, the mPTC finding was incidental. 

All patients underwent surgical therapy, which was total thyroidectomy per primam in six cases (40%) of cases, and in five patients (33.3%) of the cases, a completion to total thyroidectomy was performed after the primary lobectomy, with a median duration between the two surgeries of 93 days; for the rest of cases, in four cases (26.6%), the surgical therapy was subtotal or partial thyroidectomy. The thyroidectomy was supplemented with dissection of the cervical lymph nodes, in a single case (5%) at level III, IV, VI unilateral (right) at the first surgical intervention and levels IIa four months later.

As postoperative complications, there were five cases (33.3%) with secondary hypoparathyroidism. There have been no cases of recurrent larynx nerve injury.

I-131 therapy was performed in 73.3% of cases (11 patients), the median of the total activity of I-131 being 108 mCi (3.97 GBq) and between 17.88–420.60 mCi (0.66–15.56 GBq). According to the staging and risk group, four patients did not receive I-131 treatment.

## 4. Discussion

In this study, we present the first evaluation of pediatric mTC diagnosed in the last two decades in Romania. While the information on the outcome of adult patients with mTC is well documented in numerous studies, the paucity of mTC makes it difficult to establish evidence-based management protocols. This is particularly important for countries in which environmental risk factors for the development of thyroid nodules and thyroid cancer such as potential previous exposure to radioactive fallout and iodine deficiency are present. The present study is relevant because it illustrates the clinical characteristics of pediatric patients with mPTC in a country where these factors may modify the risk and the outcome of these tumors. 

Our data indicate that the majority of these tumors are found incidentally and confirm their favorable outcome. Nonetheless, sporadic cases with a more aggressive histological phenotype are present and may require more aggressive treatment in order to achieve remission. 

In the adult populations, in many countries, the mTC represents a large majority of the newly diagnosed cases of TC. Among all children from our database with TC diagnosed in Romania between 2000 and 2018, only 19.4% had mTC, all of them being mPTC. In a study published in 2015, conducted in California between 1988 and 2009 [4], only 8.4% thyroid microcarcinoma were identified out of a cohort of 1,825 cases of children with TC. Several explanations are possible for the discrepancy between the prevalence numbers in adult and pediatric populations. Firstly, it is possible that thyroid malignant neoplasia is less prevalent at younger ages. Secondly, children undergo less often screening investigations and surgery for unrelated conditions such as goiter or Graves’ disease, probably leading to less incidental findings. Thirdly, it is possible that the pathogenesis of TC differs in children and adults since different driver genetic events have been described in pediatric and adult TC, possibly leading to differences in presentation and outcome. Moreover, environmental factors and endocrine disruptors may be incriminated as well in the pathogenesis of TC in children, potentially to a different extent than in adults [20]. The majority of cases of mPTC were recorded in children older than 14 years old (12 cases), 80% of them were females, which is in accordance with studies from the literature [4,21,22].

Notably, in the majority of patients in our cohort, the diagnosis of mPTC was based on incidental findings at the surgery for unrelated conditions. In general, aggressive pathological features and lymph node metastases are rare in mPTC in the adult population. Nonetheless, in the present cohort, we have identified two cases that showed lymph node invasion and/or extension beyond the thyroid capsule (T3/N1b), one of which was also the only patient that underwent lymph node dissection. No patient presented distant metastases. Although given the small size of our cohort, it is not possible to estimate the true prevalence of this potentially more unfavorable phenotype, these data indicate that a careful evaluation is required preoperatively in children, even in the children that undergo surgery for a perceived benign condition. It is ideal that the preoperative ultrasound examination of the area be performed by an experienced pediatric radiologist; in this way, all the children are evaluated with a standardized protocol. In all cases, the therapeutic approach should be made by medical staff including a pediatric endocrinologist, nuclear medicine physician, pediatric surgeon, pediatrician, and pediatric radiologist. The multidisciplinary team is mandatory to be involved in these cases; the thyroid pediatric pathology is highly recommended to be handle by a specialized pediatric endocrinologist.

When the diagnosis of mTC is made in children and adolescents, numerous questions arise with respect to the most appropriate treatment. Total removal of the thyroid gland has been reported to be associated with a higher risk of postoperative complications in children than in adults. Moreover, surgery is followed by a lifelong requirement for thyroid hormone substitution. Additional treatment with I-131 has also been associated with short and long-term complications. Due to the lack of large series available including pediatric mTC patients, insufficient data are available on the outcome of these young patients treated by less aggressive approaches, such as lobectomy only. Last but not least, professional expertise with respect to treatment, follow-up, and counseling of children with TC is often scattered due to the rarity of the condition. These factors can significantly impact the long-term outcomes and the quality of life of the survivors [18]. 

Since in the large majority of patients of our study, the diagnosis was not made preoperatively, the therapeutic approach was heterogeneous, largely reflecting the real-life situation. In this study, 73.3% of cases underwent a total thyroidectomy, more than half of which were performed per primum for a different indication. Therapy was followed by the administration of I-131 in 70% of cases. The I-131 activity administered was highly variable and, in some cases, quite high, as related to the current practice, with a median of 108 mCi (3.97 GBq), ranging 17.88–420.60 mCi (0.66–15.56 GBq). I-131 activity and the number of administrations were strongly dependent on the historic period, the local available protocols, and preferences. 

The complete remission was achieved in 12 patients, 2 patients were with biochemical incomplete remission, and 1 case was with an indeterminate response. Among the patients with complete remission, four cases did not receive I-131 (26.6%), this fact underlining the excellent prognosis of the pathology. Despite the limited number of patients, the above-mentioned results necessitate conducting large, multicenter studies in order to establish the most adequate therapeutic strategy. 

The evolution was favorable with all of the patients, including the two cases with initial ATA intermediate-risk mPTC being alive, and with none of the patients having a structural disease at the last follow-up.

## 5. Limitations of the study

We should mention a number of limitations of this study. Firstly, the data were collected retrospectively. Secondly, cases were investigated and treated over a long period and in different centers; thus, there is a heterogeneous therapeutic approach, most probably explained by the different criteria of patient selection and different surgical and therapeutic protocols available at that time. Thirdly, having no pediatric patient electronic record or a national TC registry, collecting these data was difficult, leading to missing information. Last but not least, given the rarity of these pathologies, the cohort is small, resulting in a lack of statistical power, particularly for the subgroup analyses.

## 6. Conclusions

In children, TC is a rare endocrinological disease; however, the long-term consequences for the survivors are not negligible. The mPTC represented roughly one-fifth of our nationwide pediatric population diagnosed in the last 20 years. Although patients with apparently more unfavorable local phenotype were identified, this was not reflected in the outcome of the patients in terms of remission of the disease and survival. Our study illustrates the heterogeneity of the real-life practice with respect to the pediatric mPTC and underscores the need for carefully designed multicenter international studies, including larger cohorts of patients in order to provide the data required for establishing evidence-based uniform protocols. The European Reference Networks (ERN), such as the ERN for Rare Endocrine Diseases (Endo–ERN) provides an ideal platform to initiate such collaborative studies. 

## Figures and Tables

**Figure 1 children-08-00422-f001:**
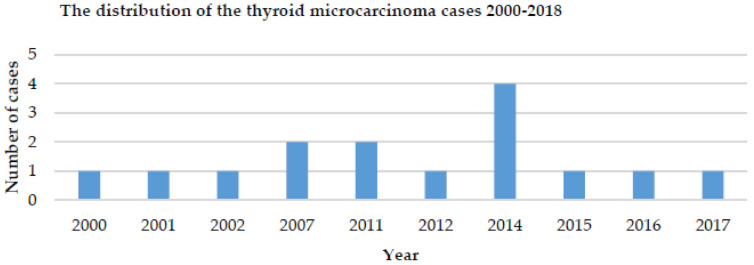
Distribution of thyroid microcarcinoma cases in children/year.

**Table 1 children-08-00422-t001:** ATA pediatric group risk [11].

ATA Pediatric Low-Risk	ATA Pediatric Intermediate-Risk	ATA Pediatric High-Risk
“Disease grossly confined to the thyroid with N0 or NX disease or patients with incidental N1a metastasis in which “incidental” is defined as the presence of microscopic metastasis to a small number of central neck lymph nodes.”	“Extensive N1a or minimal N1b disease. The impact of the pathologic identification of microscopic (ETE) (T3 disease) on management and outcomes has not been well studied in children with PTC, but patients with minimal ETE are probably either ATA Pediatric Low- or Intermediate-Risk, depending on other clinical factors.”	“Regionally extensive disease (extensive N1b) or locally invasive disease (T4 tumors), with or without distant metastasis. Patients in this group are at the highest risk for incomplete resection, persistent disease, and distant metastasis.”

**Table 2 children-08-00422-t002:** Outcome of the individual patients with mPTC.

ID.Sex, Age at Diagnosis (Years)	Follow-Up (Years)	Histology and Histological Subtype. Tumor Size (cm). TNM ^1^	Initial Treatment	Total I-131 Activity (mCi)	Outcome at Last Follow-Up
1. F, 14.6	4.1	PTC (conventional), 0.1–0.6 cm, T1amN0M0	TT, I-131	17.88	CR
2. F, 16.5	6.3	PTC (follicular variant), 1 cm, T1aN0M0	TT, I-131	64.75	CR
3. F, 15.3	3.6	PTC (conventional), 0.6 cm, T3bNxMx	TT, I-131	95.00	CR
4. M, 16.6	2.3	PTC (conventional), 0.1–1 cm T3bmN1bM0	TT, LND, I-131	107.91	CR
5. F, 12.2	16.4	PTC (follicular variant), ˂1 cm, T1amN0M0	STT, I-131	140.00	CR
6. F, 14.4	4.7	PTC (follicular variant), 0.3 cm, T1amNxM0	STT		CR
7. F, 16.8	3	PTC (conventional), 0.15 cm, T1aNxM0	TT		CR
8. M, 16.8	10.8	PTC (conventional), ˂1 cm, T1aN0M0	STT		CR
9. F, 14	4.9	PTC (follicular variant), 0.4 cm, T1amNxMx	TT, I-131	170.70	CR
10. F, 16.2	13.2	PTC (follicular variant), 0.7–1 cm, T1amN0M0	TT, I-131	311.50	CR
11. F, 17.5	1.2	PTC (diffuse sclerosing), 0.1 cm, T1amNxMx	TT, I-131	420.60	CR
12. F, 17.3	0.9	PTC (follicular variant), 0.6 cm, T1aN0M0	TT		CR
13. F, 10.5	7.6	PTC ^2^ (cystic variant), 0.4 cm, T1aNxM0	TT, I-131	50.00	BI
14. M, 15.2	8.8	PTC (conventional), 0.2–0.3 cm, T1amN0M0	STT, I-131	100.00	BI
15. F, 18.1	0.9	PTC (diffuse sclerosing), 0.5–0.6 cm, T1aNxMx	TT, I-131	150.00	IR ^3^

PTC—papillary thyroid carcinoma, TT—total thyroidectomy, STT- subtotal or partial thyroidectomy, LND—lymph node dissection. CR complete remission, BI biochemical incomplete, IR indeterminate response. ^1^ Eighth edition of 2017 TNM classification, m—multifocal tumor. ^2^ Cystic papillary in thyreoglosus cyst. ^3^ The patient had two I-131 administrations, and the last one was at the last medical visit when the Tg and antiTg were positive.

## Data Availability

The data presented in this study are available by request from the corresponding author.

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
