# Peer review of "Thyroid Microcarcinoma in Pediatric Population in Romania"

_children, 2021, doi:10.3390/children8050422_

Round 1
Reviewer 1 Report
The manuscript is interesting and concerns a topic of scientific interest and social impact. However, the manuscript has some biases. First of all, it is a retrospective work, the casuistry is limited and concerns too long a period of time with different selection centers and different approaches that ultimately determine biases that significantly alter the results due to the heterogeneity of the data. Since the casuistry is small, the statistical analysis is also distorted.
Author Response
We highly appreciate you suggestions. This study has, indeed, some biases especially because we collected the data from different centers but our intention was to analyze the data at national scale or, at least, to collect the majority of the cases from Romania so this was the only way. We do not have a national registry of pediatric thyroid cancer. We hope to do this as soon as possible, this paper being important to notice the importance of establishing one. The number of cases is small so the statistics isnˋt relevant, this is the reason why we focused on the description of the data. Despite this, the very limited number of this pathology worldwide, imposes to advertise about any similar experiences.
This is the first analysis in Romania in pediatric microcarcinoma and we hope that this is only the start and we wish to have more complex studies in the future.
Thank you!
...

Reviewer 2 Report
This manuscript by Andreea-Ioana Stefan et al describes Thyroid microcarcinoma in pediatric population in Romania. The authors indicate that diagnosis and treatment of thyroid microcarcinoma in pediatric population. However, it is unclear the detail of clinical information.
Table2 indicate that age at diagnosis and follow-up(years). However, it is difficult understand clinical information.
- Age at diagnosis by ultrasonography.
- Age at diagnosis of cancer
- Follow up (years)
- Age at surgery
Histology is important for understanding of cancer. The authors should describe histological diagnosis (Classical, Follicular variant, Solid-tubular etc..).
Tumor size is important. The authors should describe tumor size (cm or mm).
Author Response
Thank you for your suggestion and comments. It can be confusing the terminology, I agree with you. About the tabel 2, ”the age at diagnosis”, this variable is considered, as we mention in line 68-69, the age of the patient at the date of pathology report release, so after the first surgical intervention is made. ”The follow up” is considered, as we wrote in line 69-70, as the time elapsed from the date of diagnosis to the last medical visit noted in the medical file (years).
Your suggestion about the histological subtype and the tumor size is really important so we have added those details in Table 2.
Thank you for helping us to improve this manuscript.

Reviewer 3 Report
This is a retrospective observational study based on the medical records of children aged between 0 and 18 years and 6 months diagnosed with microcarcinoma from 01.01.2000-31.12.2018 in the Institute of Oncology Cluj-Napoca (IOCN) and Prof Dr. Alexandru Trestioreanu Institute of Oncology Bucharest (IOB).
Line 59 refers data was collected from 3 institutions but no third institution was noted.
Differentiated Thyroid Cancer (papillary and follicular) were identified in a total of 77 patients. 15 patients were microcarcinomas. Information in regard to clinical presentation was obtained for 10 of the 15 cases.
Line 139 “noticed by the patient of his/her parents” (maybe a mistype – please change)
Line 143 “All patients had a preoperative thyroid ultrasound (US) but only nine cases of US records were available for reviewing.”
All patients (15) included in this study were PTC histologically and I would suggest that “papillary” should be mentioned in Introduction and all over the paper instead of mTC (thyroid microcarcinoma) but mPTC.
Specific guidelines are needed to be created for children with thyroid cancer and this paper is a start but a lot more research is required on this field.
Limitations are understood and I appreciate the authors’ attention to the fact that this is a very small group with missing information. and consequently lack of statistical analysis. If tissue from these cases are available, including the other 62 pediatric non-medullary Thyroid Cancer it would be very interesting to add genetic studies and explore pathogenesis of thyroid cancer in children.
Author Response
Thank you for your attention, your comments and your appreciation.
Here are the responses point-by-point:
- we corrected the suggested errors.
- About the line 59, there was a typing error. We corrected this. We have made the changes about the papillary thyroid microcarcinoma.
- This study it is, indeed, a very small one but we hope that this is only the start of future research. We really need specific guidelines for the pediatric patients with thyroid cancer including thyroid microcarcinoma.
- It will be very interesting, we agree, to have genetic studies of thyroid cancer in children, but since the study includes a long period, there were consistent changes and evolving informations. We have information that some recent tissues from the cases are available.
With respect and consideration,

Round 2
Reviewer 1 Report
The work carried out by the authors is appreciable but in my opinion
the manuscript presents a structural approach with biases that cause
a significant alteration of the results. In my opinion, there remains
the limit of the retrospective study and the heterogeneity of the data
collected.
Author Response
Thank you for your attention, your comments and your appreciation.
Here are the responses point-by-point:
- we corrected the suggested errors.
- about the age stratification, your comments are useful and very pertinent. Yes, puberty may start at a younger age in children. Considering this, we cannot conclude that puberty status may be a predisposing factor in the development of TC but we can describe the data. The majority of mPTC cases were at adolescents over 14 years old, predominantly in girls, those data are accordingly with data from the literature. We would like to mention these details, especially because in 2015 - ATA guidelines this is a rating B recommendation.
- this is a descriptive study so we removed the statistical analysis form the methods section accordingly
- we introduced the suggested comments about the imperative role of specialized endocrinologist pediatrician in the evaluation of this children.
Thank you for your suggestions and for your help in improving this manuscript.
With respect and consideration,
In the name of all co-authors,
Andreea Stefan

Reviewer 3 Report
I appreciate the feedback from the authors. Indeed, his a a very small cohort and hopefully a start to future research leading to a much needed specific guidelines for the pediatric patients. The improvement in its content and presentation is, at this moment, insufficient for publication.
Author Response
Thank you for your attention, your comments and your appreciation.
Here are the responses point-by-point:
- we corrected the suggested errors.
- Indeed this is a descriptive study so we removed the statistical analysis form the methods section accordingly. The limited number of cases is the main concern, but as you mentioned we consider imperative to continue to share every experience in order to contribute in the future for establishing a protocol for these cases.
Furthermore, we introduced the suggested comments about the imperative role of specialized endocrinologist pediatrician in the evaluation of this children. In order to provide the best care for this pathology is mandatory to involve the pediatric endocrinologists in this network. The limited number of specialists or even the complete lack of these physicians in some countries, should be a very important point of attention for everyone.
Thank you for your suggestions and for your help in improving this manuscript.
With respect and consideration,
In the name of all co-authors,
Andreea Stefan
